# scNMT-seq enables joint profiling of chromatin accessibility DNA methylation and transcription in single cells

Stephen J. Clark [1], Ricard Argelaguet[2,3], Chantriolnt-Andreas Kapourani [4] Thomas M. Stubbs[1], Heather J. Lee[1,5,6], Celia Alda-Catalinas [1], Felix Krueger [7] Guido Sanguinetti[4], Gavin Kelsey [1,8] John C. Marioni [2,3,5] Oliver Stegle [2] Wolf Reik[1,5,8]

Parallel single-cell sequencing protocols represent powerful methods for investigating regulatory relationships, including epigenome-transcriptome interactions. Here, we report a single-cell method for parallel chromatin accessibility, DNA methylation and transcriptome profiling. scNMT-seq (single-cell nucleosome, methylation and transcription sequencing) uses a GpC methyltransferase to label open chromatin followed by bisulfite and RNA sequencing. We validate scNMT-seq by applying it to differentiating mouse embryonic stem cells, finding links between all three molecular layers and revealing dynamic coupling between epigenomic layers during differentiation.

[1] Epigenetics Programme, Babraham Institute, Cambridge CB22 3AT, UK. [2] European Molecular Biology Laboratory, European Bioinformatics Institute, Hinxton, Cambridge CB10 1SD, UK. [3] Cancer Research UK Cambridge Institute, University of Cambridge, Cambridge CB2 0RE, UK. [4] School of Informatics, University of Edinburgh, Scotland EH8 9AB, UK. [5] Wellcome Trust Sanger Institute, Hinxton, Cambridge CB10 1SA, UK. [6] School of Biomedical Sciences and Pharmacy, The University of Newcastle, Callaghan, NSW 2308, Australia. [7] Bioinformatics Group, Babraham Institute, Cambridge CB22 3AT, UK. [8] Centre for Trophoblast Research, University of Cambridge, Cambridge CB2 3EG, UK. These authors contributed equally: Stephen J. Clark and Ricard Argelaguet. These authors jointly supervised this work: Oliver Stegle and Wolf Reik. Correspondence and requests for materials should be addressed to S.J.C. (email: stephen.clark@babraham.ac.uk) or to J.C.M. (email: marioni@ebi.ac.uk) or to O.S. (email: oliver.stegle@ebi.ac.uk) or to W.R. (email: wolf.reik@babraham.ac.uk)

Understanding regulatory associations between the epigenome and the transcriptome requires simultaneous profiling of multiple molecular layers. Previously, such multi-omics analyses have been limited to bulk assays, which profile ensembles of cells. These methods have been applied to study variation across individuals[1], cell type[2] or conditions by assessing links between different molecular layers. With rapid advances in single-cell technologies, it is now possible to leverage variation between single cells to probe regulatory associations within and between molecular layers. For example, we and others have established protocols that allow the methylome and the transcriptome or, alternatively, the methylome and chromatin accessibility to be assayed in the same cell[3–7]. However, it is well known that DNA methylation and other epigenomic layers, including chromatin accessibility, do not act independently of one another[8]. Consequently, the ability to profile, at single cell resolution, multiple epigenetic features in conjunction with gene expression will be critical for obtaining a more complete understanding of epigenetic dependencies and their associations with transcription and cell states[9].

To address this, we have developed a method that enables the joint analysis of the transcriptome, the methylome and chromatin accessibility. Our approach builds on previous parallel protocols such as single-cell methylation and transcriptome sequencing (scM&T-seq[3]), in which physical separation of DNA and RNA is performed prior to a bisulfite conversion step and the cell's transcriptome is profiled using a conventional Smartseq2 protocol[10]. To measure chromatin accessibility together with DNA methylation, we adapted Nucleosome Occupancy and Methylation sequencing (NOMe-seq)[11], where a methyltransferase is used to label accessible (or nucleosome depleted) DNA prior to bisulfite sequencing (BS-seq), which distinguishes between the two epigenetic states. In mammalian cells, cytosine residues in CpG dinucleotides can be abundantly methylated, whereas cytosines followed by either adenine, cytosine or thymine (collectively termed CpH) are methylated at a much lower rate[12]. Consequently, by using a GpC methyltransferase (M.CviPI) to label accessible chromatin, NOMe-seq can recover endogenous CpG methylation information in parallel. NOMe-seq is particularly attractive for single-cell applications since, contrary to count-based assays such as ATAC-seq or DNase-seq, the GpC accessibility is encoded through the bisulfite conversion and hence inaccessible chromatin can be directly discriminated from missing data. Importantly, this implies that the coverage is not influenced by the overall accessibility, so lowly accessible sites will not suffer from increased technical variation compared to highly accessible sites. Additionally, the resolution of the method is determined by the frequency of GpC sites within the genome (~1 in 16 bp), rather than the size of a library fragment (>100 bp). Recently developed single-cell NOMe-seq protocols have been applied to assess cell-to-cell variance in CTCF footprinting[6] and to map chromatin remodelling during preimplantation development[7]. However, no method that combines RNA-seq with chromatin accessibility profiling in the same cells (with or without DNA methylation) has been reported to-date, which is critical for studying interactions between the epigenome and the transcriptome.

## Results

**scNMT-seq robustly profiles each molecular layer.** To validate scNMT-seq, we applied the method to a batch of 70 serum-grown EL16 mouse embryonic stem cells (ESCs), together with four negative (empty wells) and three scM&T-seq controls (cells processed using scM&T-seq, i.e., without M.CviPI enzyme treatment). This facilitates direct comparison with previous methods for assaying DNA methylation and transcription in the same cell[3,13], as well as providing a control of bisulfite conversion efficiency within the experiment. We isolated cells into methyltransferase reaction mixtures using FACS, followed by the physical separation of the DNA and RNA prior to BS-seq and RNA-seq library preparation (see Fig. 1a for an illustration of the protocol). Alignment of the BS-seq data and other bioinformatics processing can be carried out using established pipelines, with the addition of a filter to discard G–C–G positions, for which it is intrinsically not possible to distinguish endogenous methylation from in vitro methylated bases (21% of CpGs genome-wide). Similarly, we discard C–C–G positions to mitigate against possible off-target effects of the enzyme[11] (27% of CpGs). In total, 61 out of 70 cells processed using scNMT-seq passed quality control for both BS-seq and RNA-seq (Methods, Supplementary Data 1).

The requirement to filter out C–C–G and G–C–G positions from the methylation data reduces the number of genome-wide cytosines that can be assayed from 22 million to 11 million. However, despite this, a large proportion of genomic loci with regulatory roles, such as promoters and enhancers, can in principle be assessed by scNMT-seq (Fig. 1b). Consistent with this, we observed high empirical coverage for methylation: a median of ~50% of promoters, ~75% of gene bodies and ~25% of active enhancers are captured in a typical cell by at least 5 cytosines (Fig. 1c, Supplementary Fig. 1a). We also compared the methylation coverage to data from our previous BS-seq protocols that did not incorporate a DNA accessibility component[3], again finding only small differences in coverage, albeit these became more pronounced when down-sampling the total sequence coverage (Supplementary Fig. 1b). Computational methods for imputing these missing values could help to further mitigate these differences[14]. Due to the higher frequency of GpC compared to CpG dinucleotides in the mouse genome, accessibility coverage was larger than that observed for endogenous methylation (Fig. 1b, c and Supplementary Fig. 1a). Using our data, a median of ~85% of gene bodies and ~75% of promoters could be probed for DNA accessibility, the highest coverage achieved by any single-cell accessibility protocol to date (9.4% using scATAC-seq[15], and with scDNase-seq, ~50% of genes >1 RPKM, >80% of genes >3 RPKM[16]). This coverage also compares favourably with other single-cell NOMe-seq methods developed in parallel, which report GpC site coverages of 2.9%[6] and 10%[7] compared to 15% using scNMT-seq (Supplementary Data 1).

Next, we examined accessibility levels at loci with known regulatory roles. We found that accessibility was increased at known DNaseI hypersensitivity sites, super enhancer regions and binding sites for transcription factors and other DNA binding proteins (from published ChIP-seq data, Fig. 1d, Supplementary Fig. 2). As, a control, we included cells which did not receive enzyme treatment (scM&T-seq controls) and these cells showed universally low GpC methylation levels (~2%), with no enrichment at regulatory regions, indicating that the accessibility data are not affected by endogenous GpC methylation (Supplementary Fig. 3). We next stratified loci and cells based on the expression level of the nearest gene (based on the RNA data from the corresponding cell). In agreement with previous studies[8], we observed that highly-expressed genes were associated with increased accessibility at promoters and at nearby regulatory sites, whereas lowly-expressed genes were associated with reduced accessibility (Fig. 1e; Supplementary Fig. 4).

Next, to assess the quality of data obtained using scNMT-seq, we compared the transcriptome, methylome and accessibility profiles to published datasets. When considering the RNA-seq component, dimensionality reduction[17] and hierarchical clustering revealed that cells cluster by condition and not by protocol (Supplementary Fig. 5). We next compared the methylome

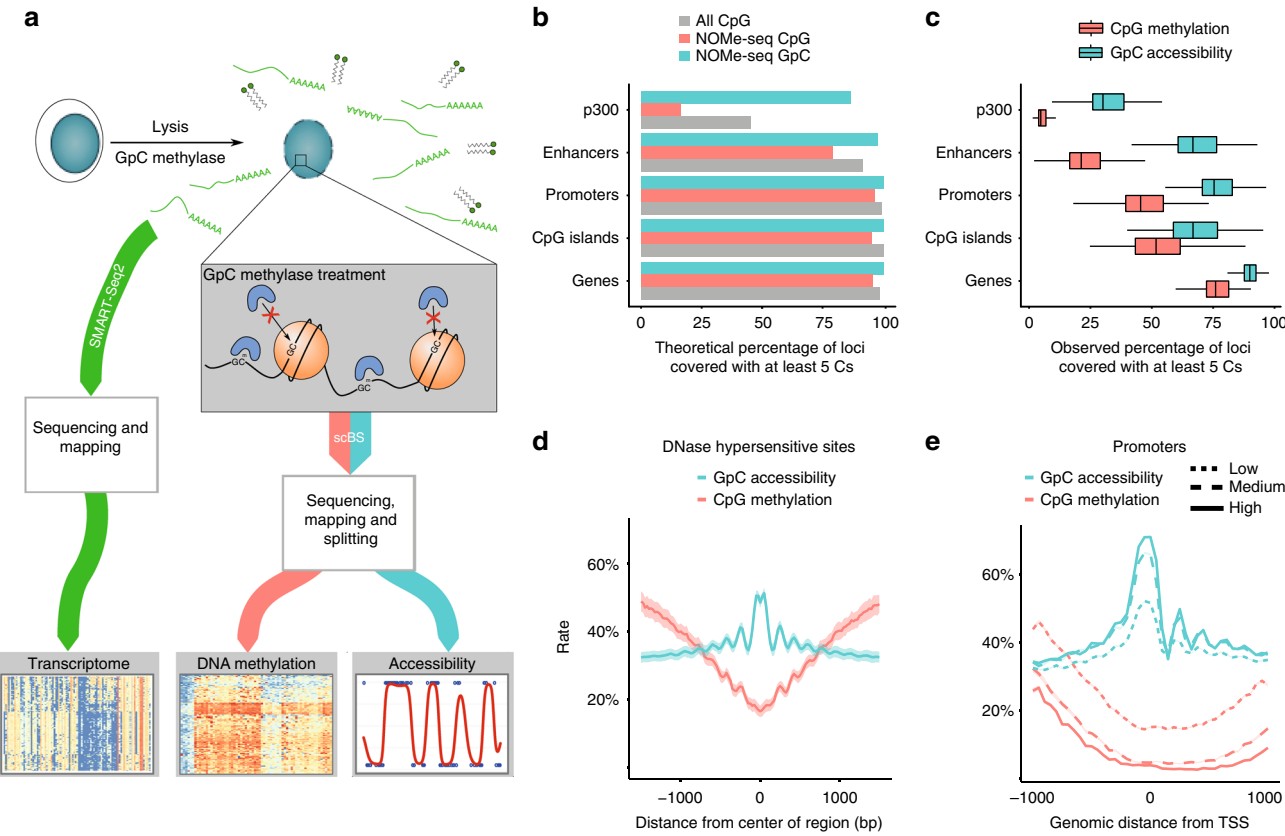

**Fig. 1** scNMT-seq overview and genome-wide coverage. **a** Protocol overview. Single-cells are lysed and accessible DNA is labelled using GpC methyltransferase. RNA is then separated and sequenced using Smart-seq2, whilst DNA undergoes scBS-seq library preparation and sequencing. Methylation and chromatin accessibility data are separated bioinformatically. **b** Theoretical maximum CpG coverage of genomic contexts with known regulatory roles. Shown is the proportion of loci in different contexts that contain at least 5 cytosines. 'All CpG' considers any C-G dinucleotides (e.g., as in scBS-seq), 'NOMe-seq CpG' considers A-C-G and T-C-G trinucleotides and 'NOMe-seq GpC' considers G-C-A, G-C-C and G-C-T trinucleotides. **c** Empirical coverage in 61 mouse ES cells considering the same contexts as in **b**. Shown is the coverage distribution across cells after QC; box plots show median coverage and the first and third quartile, whiskers show 1.5 × the interquartile range above and below the box. **d** CpG methylation and GpC accessibility profiles at published DNase hypersensitive sites[19]. The profiles were computed as a running average in 50 bp windows. Shading denotes standard deviation across cells. **e** CpG methylation and GpC accessibility profiles at gene promoters. Promoters are stratified by average expression level of the corresponding gene (log normalised counts less than 2 (low), between 2 and 6 (medium) and higher than 6 (high). The profile is generated by computing a running average in 50 bp windows

obtained from scNMT-seq to single-cell libraries profiled using scM&T-seq[3], scBS-seq[13] and bulk BS-seq[18], finding that most of the cell-to-cell variation is not attributed to protocol or study but to changes in the mean methylation rate (first principal component, 51% variance) (Supplementary Fig. 6). To validate the accessibility measurements, we generated a synthetic pseudo-bulk dataset by merging data from all cells, which we compared to published bulk DNase-seq data from the same cell type[19]. Globally, we observed high consistency between datasets (Relative accessibility profiles, Pearson $R = 0.74$, Supplementary Fig. 7). A notable difference was that scNMT-seq data captured, within single cells, oscillating profiles with peaks spaced ~180 to ~200 bp apart, indicating the positions of nucleosomes (Fig. 1d, e and Supplementary Fig. 8), which is consistent with accessibility profiles obtained using bulk NOMe-seq[11], demonstrating high resolution of our accessibility measurements.

As a final quality assessment, we analysed associations between molecular layers within individual cells (across all genes), which is similar to approaches used to investigate linkages using bulk data (see Fig. 2 upper panel for a graphical representation). Reassuringly, this confirmed the expected negative correlations for methylation with transcription[12] and methylation with

accessibility[8] (Fig. 2, lower panel) and positive correlations between accessibility and expression[18] (for most genomic contexts with the notable exception of active enhancers for which there is little evidence for a correlation between accessibility and expression in our data or in published data). These results indicate that our method recapitulates, within single cells, known trends from bulk data.

Taken together, these results demonstrate that our method is able to robustly profile gene expression, DNA methylation and chromatin accessibility within the same single cell.

**Identifying genomic loci with coordinated variability**. Having established the efficacy of our method, we next explored its potential for identifying loci with coordinated epigenetic and transcriptional heterogeneity. To obtain a dataset with a larger degree of heterogeneity than observed in ES cells, we prepared a second dataset obtained from serum grown ES cells that we removed from LIF for 3 days to initiate differentiation into embryoid bodies (EBs). We sequenced 43 cells, which clearly clustered into two sub-populations based on RNA-seq profiles, corresponding to pluripotent and differentiating states (Supplementary Fig. 9). First, we examined cell-to-cell variance in the

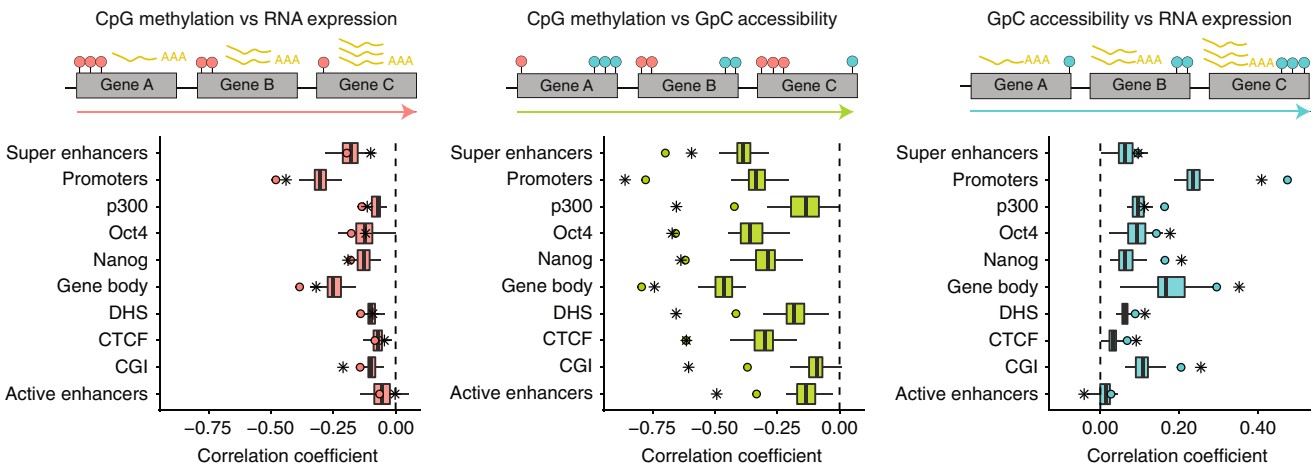

**Fig. 2** scNMT-seq recapitulates known global associations between molecular layers. Upper panel shows an illustration of the computation of the correlation across genes (one association test per cell). Left is CpG methylation and RNA expression associations, middle is CpG methylation and GpC accessibility associations, and right is GpC accessibility and RNA expression associations. Red circles represent CpG methylation levels, blue circles represent GpC accessibility levels and yellow polyA tails represent RNA abundance. Lower panel shows the Pearson correlation coefficients between molecular layers at different genomic contexts in the ESC data. Box plots show the distribution of correlation coefficients in single cells. Boxes display median coverage and the first and third quartile, whiskers show 1.5 × the interquartile range above and below the box. Dots show the correlation coefficient in the pseudo-bulked data estimated as average across all single-cells. Stars show the correlation coefficient using published bulk data from the same cell type[18,19]. Sample size for the single-cell data is determined by the number of cells which pass QC for both layers (61–64 cells, see Methods)

methylation data, finding that enhancers and Nanog binding sites were associated with the largest methylation heterogeneity, which is in agreement with previous ES cell data[3,13] (Supplementary Fig. 10a, b). Conversely, variability in accessibility rates was either at similar levels to the background or, in the case of promoters, CGIs, active enhancers, and gene bodies, found to be reduced relative to the background (Supplementary Fig. 10c, d). This could indicate that there are genomic elements which limit variability of chromatin accessibility, such as CGIs most of which in a cell are constitutively accessible[20].

Subsequently, we tested locus-specific associations between different pairwise combinations of molecular layers (Fig. 3a), which is distinct from the correlations across genes used for quality control above and is enabled by parallel single-cell measurements in multiple cells. This analysis can be used to discover individual genes and loci with coordinated heterogeneity across pairs of molecular layers. First, considering associations between methylation and transcription, we identified a minimum of 3 (exons) and a maximum of 47 (gene bodies) associations (FDR <0.1, Fig. 3a, Supplementary Fig. 11a, Supplementary Data 2, Methods). The majority of these associations were negative, reflecting the known relationship between these two layers. In contrast, we found that associations between DNA accessibility and transcription were less widespread, with a small number of mostly positive associations in promoters, p300 binding sites and super enhancer regions (13 associations total, FDR <0.1, Fig. 3a, Supplementary Fig. 11b and Supplementary Data 2). Low numbers of correlated accessibility–expression could indicate that transcriptional changes in this population are more dependent on DNA methylation changes than chromatin accessibility changes and this is in agreement with the results presented in Fig. 2. Finally, for methylation-accessibility, we found associations at most genomic contexts, with up to 89 significant correlations (introns) and these tended to be negative as expected (Fig. 3a, Supplementary Fig. 11c and Supplementary Data 2).

As an illustrative example, Fig. 3b displays the *Esrrb* locus, a gene we find to be expressed primarily in the pluripotent cells

(Supplementary Fig. 9), and which displays a strong correlation between methylation and expression in super enhancer regions, replicating previous findings[3]. Mean methylation and accessibility rates along the gene showed clear differences between the two sub-populations of cells identified, which were largest at regulatory elements. While the super enhancers showed the strongest negative correlation between methylation and transcription, a strong positive correlation was found in the promoter between accessibility and transcription. Similarly, Supplementary Fig. 12 shows the *Prtg* locus, a known neuroectoderm marker[21], which is expressed primarily in differentiated cells (Supplementary Fig. 9), again showing marked epigenetic differences between the two cell populations.

**Base resolution chromatin accessibility profiles.** Inspection of accessibility data at the single GpC level reveals complex patterns due to presence of nucleosomes (Fig. 1d, e), which are not appropriately captured by rate parameters calculated in pre-defined windows. The prevalence of these oscillatory patterns prompted us to reconstruct the DNA accessibility profiles in individual cells at a locus level, by adapting a statistical model initially developed for DNA methylation profiles[22]. As expected, the single-cell profiles at gene promoters were more predictive of gene expression than conventional accessibility rates (Supplementary Fig. 13), and these captured characteristic patterns of nucleosome depleted regions at transcription start sites and cell-to-cell variation in both the position and the number of nucleosomes (see Supplementary Fig. 14).

Next, we exploited the reconstructed profiles to quantify the level of heterogeneity of chromatin accessibility at transcription start sites. For each gene, we clustered the cells based on the similarity of the accessibility profiles and we estimated the most likely number of clusters (Methods). Subsequently, we stratified genes by the number of clusters estimated by our model, which we considered as a measure of accessibility heterogeneity (Fig. 4a). This revealed that genes with homogeneous accessibility profiles (fewer clusters) were associated with higher average expression levels (Fig. 4b) and were enriched for gene ontology terms linked

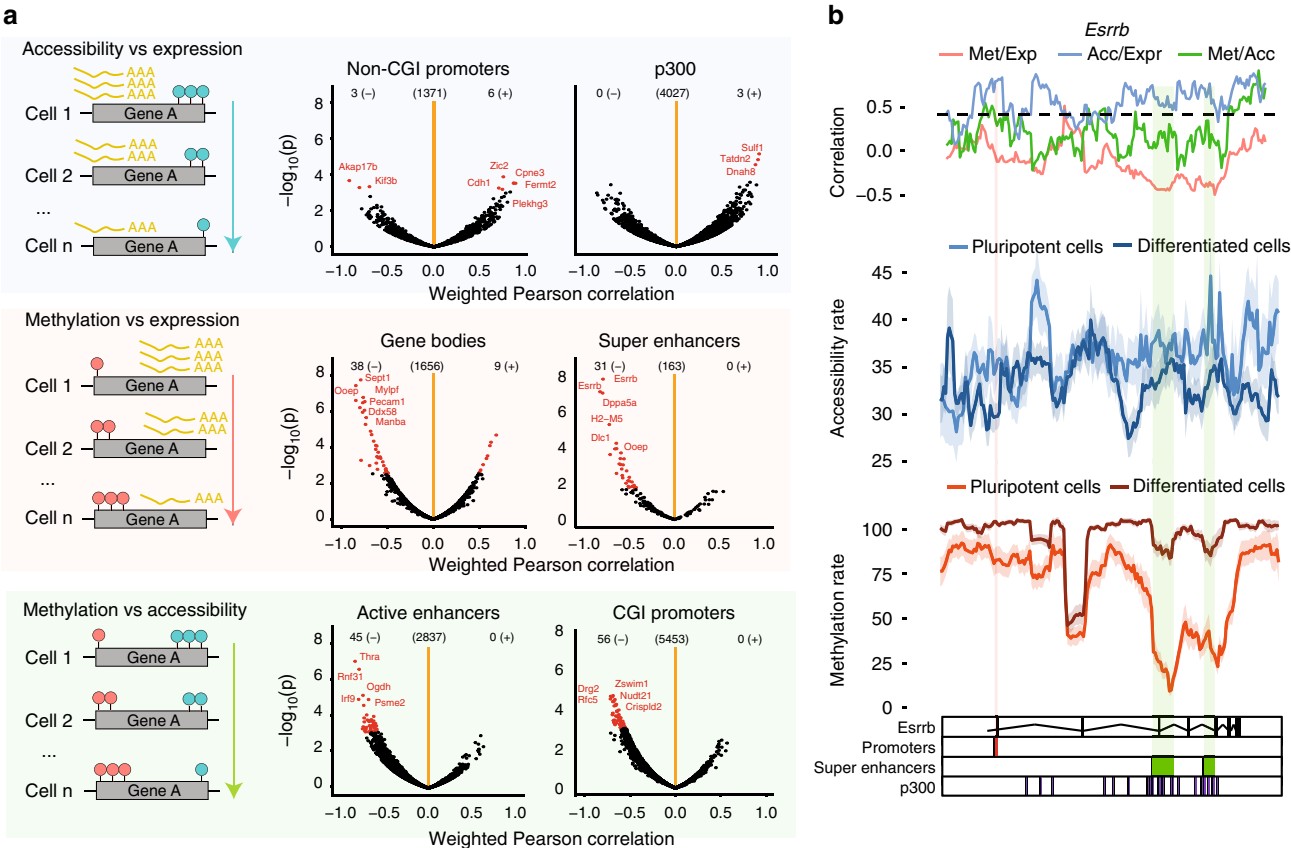

**Fig. 3** scNMT-seq enables the discovery of novel associations at individual loci. **a** Left panel shows an illustration for the correlation analysis across cells, which results in one association test per locus. The right panel shows the Pearson correlation coefficient (x-axis) and log10 p-value (y-axis) from association tests between different molecular layers at individual loci, stratified by genomic contexts. Significant associations (FDR <0.1, Benjamini–Hochberg adjusted), are highlighted in red. The number of significant positive (+) and negative (−) associations and the number of tests (centre) are indicated above. Sample size varies depending on the number of cells, which have coverage for a specific loci (see Methods). **b** Zoom-in view of the *Esrrb* gene locus. Shown from top to bottom are: Pairwise Pearson correlation coefficients between each pair of the three layers (Met methylation, Acc accessibility, Expr expression). Accessibility (blue) and methylation (red) profiles shown separately for pluripotent and differentiated sub-populations; mean rates (solid line) and standard deviation (shade) were calculated using a running window of 10 kb with a step size of 1000 bp; Track with genomic annotations, highlighting the position of regulatory elements: promoters, super enhancers, and p300 binding sites

to house-keeping functions, such as regulation of gene expression, rRNA processing and splicing (Fig. 4d). Examples of genes with a single cluster are shown in Supplementary Fig. 15 and examples of genes with two differentially expressed clusters are shown in Supplementary Fig. 16. In contrast, genes with heterogeneous accessibility (multiple clusters) were associated with low expression levels and were enriched for bivalent promoters containing both active H3K4me3 and repressive H3K27me3 histone marks (Fig. 4c). The increased bivalency was independent of the mean expression level of the gene (Supplementary Fig. 17).

**Epigenome dynamics along a developmental trajectory**. One of the most interesting opportunities of scNMT-seq is to link epigenetic properties to the transcriptomic profile along dynamic trajectories of different cell states. To explore this, we used the RNA-seq component to reconstruct a pseudotemporal ordering of the cells from pluripotent to differentiated cell states (Fig. 5a, Methods). We then tested for coordinated changes between the accessibility profiles and the cellular position in the differentiation trajectory, which identified a set of 15 genes that showed a coherent dynamic pattern (Supplementary Fig. 18, Methods). Fig. 5b depicts two representative genes: *Efhd1*, a gene that displays a transition from a state with an open transcription start site

(TSS) to a state with a closed TSS; and *Rock2*, with a similar transition on the +1 nucleosome after the TSS. Supplementary Fig. 19 shows additional examples of genes with associations between accessibility profile and pseudotime trajectory.

Finally, we investigated whether dynamic changes in the coupling between the epigenetic layers are observed along the differentiation trajectory. To this end, we plotted methylation-accessibility correlation coefficients (as calculated in Fig. 2a) against pseudotime, which revealed an increasing negative correlation coefficient between DNA methylation and accessibility in practically all genomic contexts (Fig. 5c). Notably, this suggests that the coupling between the epigenetic layers increases as cells commit to downstream lineages, which could be an important step in lineage priming. Importantly, this analysis was made possible by the continuous nature of the single-cell pseudotime ordering and the ability to profile the three molecular layers and highlights the utility of such parallel single-cell techniques.

In conclusion, we have described a method for parallel single-cell DNA methylation, gene expression and high-resolution chromatin accessibility measurements and report novel associations between each molecular layer. We additionally show that our method can be used to dissect the dynamics of epigenome

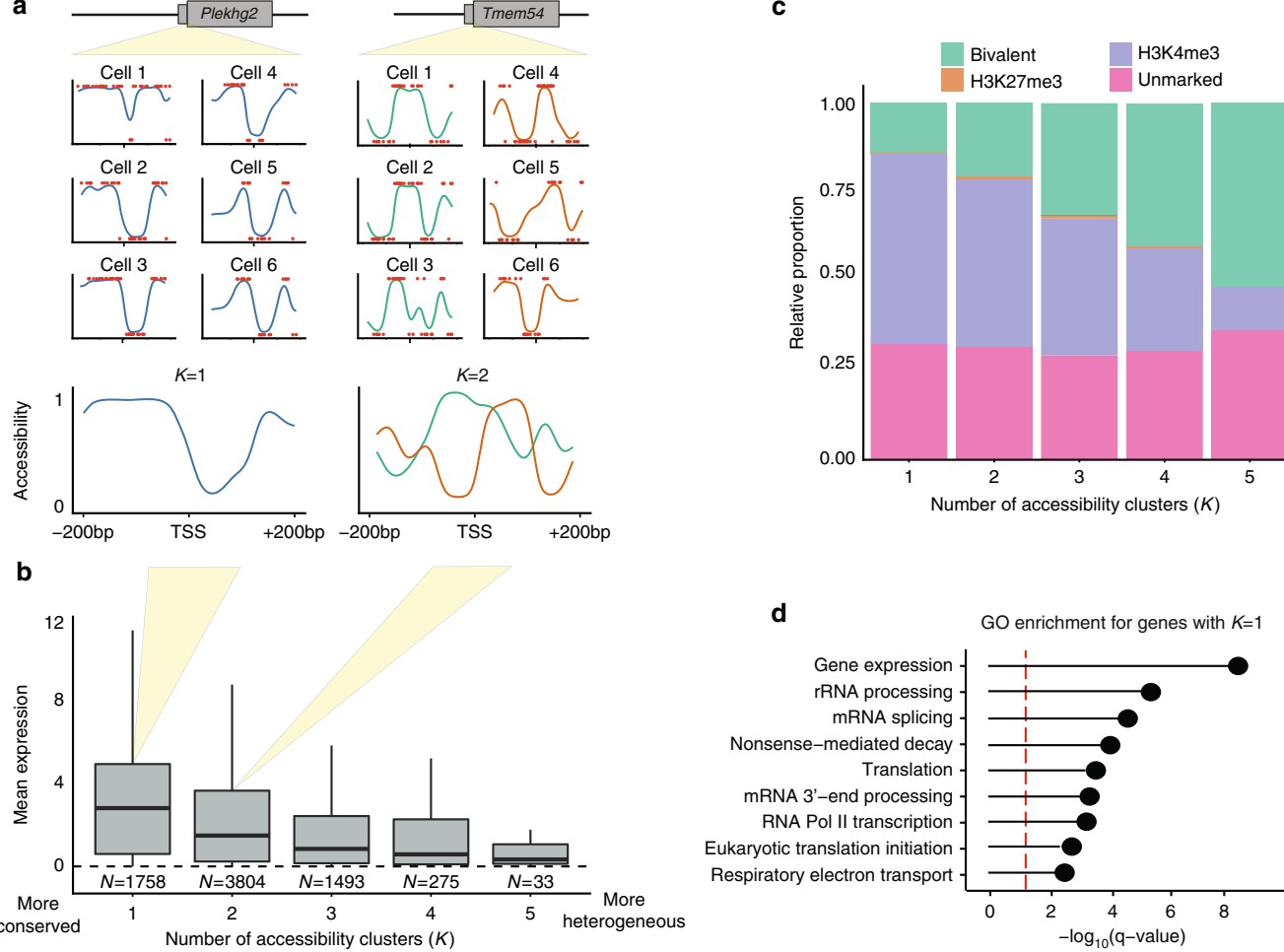

**Fig. 4** Modelling chromatin accessibility profiles at gene promoters in single cells. **a** Accessibility profiles for each cell and gene are fitted at a single nucleotide resolution ($+/-200$ bp around the TSS), followed by clustering of profiles for each gene to estimate the most likely number of clusters. Genes with higher numbers of clusters correspond to genes with increased heterogeneity compared to genes with small numbers of clusters. **b** Relationship between heterogeneity in the accessibility profile and gene expression. Boxplots show the distribution of average gene expression levels for genes with increasing numbers of accessibility clusters. Upper and lower hinges display third and first quartiles; the bar displays the median and the whiskers 1.5 times the inter-quartile range above and below the boxes. **c** Proportion of gene promoters marked with H3K4me3 and/or H3K27me3 stratified by number of accessibility clusters. Promoters with high levels of accessibility heterogeneity are associated with the presence of bivalent histone marks (both H3K4me3 and H3K27me3). **d** Gene ontology terms significantly enriched in genes with most homogeneous accessibility profiles ($K = 1$)

interactions during a developmental trajectory. This method will greatly expand our ability to investigate relationships between the epigenome and transcriptome in heterogeneous cell types and across developmental and other cell fate transitions.

## Methods

**Experimental design**. No statistical methods were used to predetermine sample size. The experiments were not randomised. The investigators were not blinded to allocation during experiments and outcome assessment.

**Cell culture**. El16 mESCs were derived from a 129xCast/129 embryo previously[23] and cultured in serum containing media (DMEM 4500 mg/l glucose, 4 mM L-glutamine, 110 mg/l sodium pyruvate, 15% foetal bovine serum, 1 U/ml penicillin, 1 µg/ml streptomycin, 0.1 mM nonessential amino acids, 50 µM β-mercaptoethanol, and 103 U/ml LIF ESGRO) without feeders. E14 mESCs (the E14 cell line was a generous gift from A. Smith) were cultured as EL16 then seeded into low attachment plates at 1000 cells mL$^{-1}$ in serum media without LIF for 3 days before collection. Single cells were collected by FACS, selecting for live cells and low DNA content (i.e., G0 or G1 phase cells) using ToPro-3 and Hoechst 33342 staining. The cell lines were subjected to routine mycoplasma testing using the MycoAlert testing kit (Lonza).

**Library preparation**. Cells were collected directly into 2.5 µl methyltransferase reaction mixture which was comprised of 1 × M.CviPI Reaction buffer (NEB), 2 U M.CviPI (NEB), 160 µM S-adenosylmethionine (NEB), 1 U µl$^{-1}$ RNAsein (Promega), 0.1% IGEPAL (Sigma) then incubated for 15 min at 37 °C. The reaction was stopped and the RNA preserved with the addition of 5 µl RLT plus (Qiagen) prior to scM&T-seq library preparation according to the published protocols for G&T-seq[24,25] and scBS-seq[26] with minor modifications. Briefly, mRNA was captured using Smart-seq2[10,27] oligo-dT pre-annealed to magnetic beads (MyOne C1, Invitrogen). The lysate containing the gDNA was transferred to a separate PCR plate and the beads were washed three times in 15 µl of 1 × FSS buffer (Superscript II, Invitrogen), 10 mM DTT, 0.005% tween-20 (Sigma) and 0.4 U µl$^{-1}$ of RNAsin (Promga). After each wash, the solution was transferred to the DNA plate to maximise recovery. The beads were then resuspended in 10 µl of reverse transcriptase mastermix (100 U SuperScript II (Invitrogen), 10 U RNAsin (Promega) 1 × Superscript II First-Strand Buffer, 2.5 mM DTT (Invitrogen), 1 M betaine (Sigma), 9 mM MgCl2 (Invitrogen), 1 µM Template-Switching Oligo[10,27] (Exiqon), 1 mM dNTP mix (Roche)) and incubated on a thermocycler for 60 min at 42 °C followed by 30 min at 50 °C and 10 min at 60 °C. PCR was then performed by adding 11 µl of 2 × KAPA HiFi HotStart ReadyMix and 1 µl of 2 µM ISPCR primer[10,27] and cycling as follows: 98 °C for 3 min, then 18 cycles of 98 °C for 15 s, 67 °C for 20 s, 72 °C for 6 min and finally 72 °C for 5 min. cDNA was purified using a 1:1 volumetric ratio of Ampure Beads (Beckman Coulter) and eluted into 20 µl of water. Libraries were prepared from 100 to 400 pg of cDNA using the Nextera XT Kit (Illumina), per the manufacturer's instructions but with one-fifth volumes.

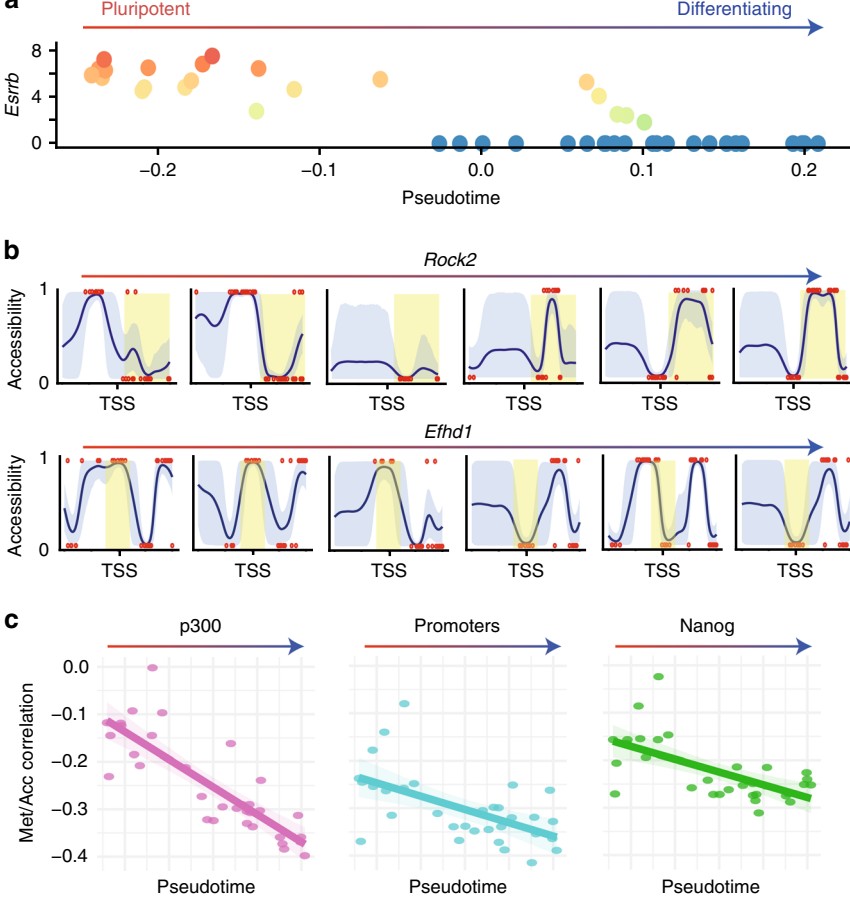

**Fig. 5** Using scNMT-seq to explore dynamics of the epigenome during differentiation. **a** Embryoid body cells ordered in a developmental trajectory inferred from the RNA-seq data. Shown is the location of each cell in pseudotime (x axis) versus the expression level of *Esrrb* (y axis). **b** Reconstructed dynamics of variation in chromatin accessibility profiles across the developmental trajectory. Shown are profiles of representative cells for *Rock2* and *Efhd1*. Axis ticks display −200 bp, 0 bp and +200 bp relative to the TSS. Shading is used to highlight changes between cells. **c** Developmental trajectory is associated changes in genome-wide methylation-accessibility coupling. Shown is the location of each cell in pseudotime (x axis) and the corresponding Pearson correlation coefficients between methylation and accessibility (y axis) in different genomic contexts

In parallel, the genomic DNA was purified with a 0.8:1 volumetric ratio of Ampure XP Beads (Beckman Coulter) and eluted into 10 μl of water. Bisulfite conversion was carried out using EZ Methylation Direct MagBead kit (Zymo) according the manufacturers' instructions but with half volumes. Converted DNA was eluted into 40 μl of first strand synthesis mastermix (1 × Blue Buffer (Enzymatics), 0.4 mM dNTP mix (Roche), 0.4μM 6NF oligo (IDT) then heated to 65 °C for 3 min and cooled on ice. 50U of klenow exo- (Enzymatics) was added and the mixture incubated on a thermocycler at 37 °C for 30 min after slowly ramping from 4 °C. First strand synthesis was repeated 4 more times with the addition of 0.25 μl of reaction mixture (1× blue buffer, 0.25 mM dNTPs, 10 mM 6NF oligo and 25U klenow exo-). Reactions were diluted to 100 μl and 20U of exonuclease I (NEB) added and incubated at 37 °C before purification using a 0.75:1 ratio of AMPure XP beads. Purified products were resuspended in 50 μl of second strand mastermix (1× Blue Buffer (Enzymatics), 0.4 mM dNTP mix (Roche), 0.4 μM 6NF oligo (IDT) then heated to 98 °C for 2 min and cooled on ice. 50U of klenow exo- (Enzymatics) was added and the mixture incubated on a thermocycler at 37 °C for 90 min after slowly ramping from 4 °C. Second strand products were purified using a 0.75:1 ratio of AMPure XP beads and resuspended in 50 μl of PCR mastermix (1× KAPA HiFi Readymix, 0.2 μM PE1.0 primer, 0.2 μM iTAG index primer) and amplified with 14 cycles. Finally, scBS-seq libraries were purified using a 0.7:1 volumetric ratio of AMPure XP beads before pooling and sequencing.

**Sequencing EL16 serum ES cells.** Twenty of the BS-seq libraries, including 3 negative controls, were initially sequenced on a 50 bp single-end MiSeq run to assess quality. The negative controls were found to have substantially reduced mapping efficiencies compared to the single cell samples (mean of 2.7% compared to 36.8%, see Supplementary Data 1). All single-cell BS-seq libraries were subsequently sequenced to a mean depth of 16.1 million paired-end reads and RNA-seq libraries were sequenced to a mean depth of 2.0 million paired-end reads. Both sets

of libraries were sequenced on HiSeq 2500 instruments using v4 reagents and 125 bp read length.

**Sequencing E14 embryoid body cells.** Forty eight BS-seq libraries were sequenced as a multiplex on one 75 bp PE high-output run on an Illumina NextSeq500 with a mean sequencing depth of 9.6 million per cell. RNA-seq libraries were sequenced on an Illumina NextSeq500 with a mean depth of 1.0 million 75 bp single-end reads per cell (Supplementary Data 1).

**BS-seq alignment.** Single-cell bisulfite libraries were processed using Bismark[28] as described[26] with the additional --*NOMe* option in the coverage2cytosine script which produces CpG report files containing only A–C–G and T–C–G positions and GpC report files containing only G–C–A, G–C–C and G–C–T positions.

**RNA-seq alignment.** Single-cell RNA-seq libraries were aligned using HiSat2[29] using options --dta --sp 1000,1000 --no-mixed --no-discordant for the paired-end ES cell libraries and --dta --sp 1000,1000 for the single-end EB cell libraries.

**Quality control-RNA-seq.** For the EL16 serum grown ES cells, we discarded cells that had (1) less than 300,000 reads mapped (2) more than 15% of total reads mapped to mitochondrial genes, (3) less than 2000 genes expressed. In total, 68 cells passed the quality control (Supplementary Fig. 20a).

For the E14 embryoid body cells, we used a lower read-depth cut-off due to the lower sequencing depth employed, discarding cells that had (1) less than 100,000 reads mapped (2) more than 15% of total reads mapped to mitochondrial genes, (3) less than 2000 genes expressed. In total, 46 cells passed the quality control (Supplementary Fig. 20b).

**Quality control–BS-seq**. For the EL16 serum grown ES cells, we discarded cells that had (1) less than 10% mapping efficiency (2) less than 500,000 CpG sites or 5,000,000 GpC sites covered. We additionally excluded one cell with unusually high CpG coverage (>5 M) and low duplication (26%) as a possible doublet. In total, 64 cells out of 73 passed the quality control (Supplementary Fig. 21a, Supplementary Data 1). All 64 cells also passed RNA-seq QC (88%) and these comprised 61 scNMT-seq cells and 3 scM&T-seq cells.

For the E14 EB cells, we again used a lower coverage cutoff due to lower sequencing depth, discarding cells that had (1) less than 10% mapping efficiency (2) less than 300,000 CpG sites covered. In total, 40 cells passed the quality control (Supplementary Fig. 21b, Supplementary Data 1), all of which also passed RNA-seq QC and comprised 33 scNMT-seq cells and 7 scM&T-seq cells.

**CpG Methylation and GpC accessibility quantification**. Following the approach of Smallwood et al[13], individual CpG or GpC sites in each cell were modelled using a binomial model where the number of successes is the number of reads that support methylation and the number of trials is the total number of reads. A CpG methylation or GpC accessibility rate for each site and cell was calculated by maximum a posteriori assuming a beta prior distribution. Subsequently, CpG methylation and GpC accessibility rates were computed for each genomic feature assuming a normal distribution across cells and accounting for differences in the standard errors of the single site estimates. See Supplementary Data 3 for details of genomic contexts used in this study.

**RNA quantification**. Gene expression counts were quantified from the mapped reads using featureCounts[30]. Gene annotations were obtained from Ensembl version 87[31]. Only protein-coding genes matching canonical chromosomes were considered. Following[32] the count data was log-transformed and size-factor adjusted based on a deconvolution approach that accounts for variation in cell size[33].

**Methylation and accessibility pseudo-bulk profiles**. Methylation and accessibility profiles were visualised by taking predefined windows around the genomic context of interest. For each cell and feature, methylation and accessibility values were averaged using running windows of 50 bp. The information from multiple cells was combined by calculating the mean and the standard deviation for each running window. Genes were split into three classes according to a histogram of the log2 normalised counts (x): Low (x < 2), Medium (2 < x < 6) and High (x > 6). All genomic features were associated to the closest gene within a 5 kb window (upstream and downstream of gene start and stop).

**Single-cell accessibility profiles**. Accessibility profiles were constructed within each cell and gene in +/−200 bp windows around the TSS (as displayed in Fig. 5b and Supplementary Figs. 14, 15 and 16) using a generalised linear model (GLM) of basis function regression coupled with a Bernoulli likelihood using BPRMeth[22]. We only considered genes that were covered in at least 40% of the cells with a minimum coverage of 10 GpC sites. Subsequently, we clustered the profiles for each gene by fitting a finite mixture model using an expectation–maximisation (EM) algorithm. We estimated the most likely number of clusters based on the Bayesian Information Criterion (BIC). The number of clusters was used as a measure of cell-to-cell variation in the accessibility profile; the rationale being that homogeneous profiles will be grouped in a single cluster, while regions with heterogeneous profiles will be assigned a higher number of clusters. Gene Ontology enrichment was performed for the different clusters using Fisher's exact test. The p-values where corrected by multiple testing using False Discovery Rate.

**Predicting expression**. To compare the performance of using accessibility rates versus profiles for predicting gene expression levels we used the same approach described in[22]. We first computed the accessibility rates and profiles for each gene and cell. Then, for each cell, we used the fitted values as input features to a regression model with the gene expression levels as the response variable. To measure the accuracy of the model we computed the Pearson's correlation coefficient between the observed and predicted expression levels (Supplementary Fig. 13a) To account for the different number of features used in the two models (i.e., rate vs profile features) we also computed the adjusted $R^2$ (Supplementary Fig. 13b)

**Correlation analysis**. For the correlation analysis across cells, genes with low expression levels and low variability were discarded, according to the rationale of independent filtering[34]. Only the top 50% of the most variable loci were considered for analysis and a minimum number of 20 cells was required to compute a correlation. A minimum coverage of 3 sites was required per feature. All genomic features were associated to the closest gene within a 10 kb window (upstream and downstream of gene start and stop). Following our previous approach[3], we tested for linear associations by computing a weighted Pearson correlation coefficient, thereby accounting for differences in the coverage between cells. When assessing correlations between GpC accessibility with CpG methylation, we used the average CpG methylation coverage as a weight. Two-tailed Student's $t$-tests were performed

to test for nonzero correlation, and $P$-values were adjusted for multiple testing for each context using the Benjamini–Hochberg procedure. For promoter annotations, we used a small window for accessibility (+/−50 bp) to focus our analysis on the transcription start site whereas for methylation we considered a larger window (+/−2 kb). This choice was informed by pseudo-bulking the single-cell data and computing the correlation between accessibility/methylation and gene expression (across genes) for small 50 bp windows along the promoter, finding that the strongest signal fell within our chosen range (Supplementary Fig. 22).

**Pseudotemporal ordering of cells**. Cells were ordered along a putative developmental trajectory (pseudotime) with the destiny package[35], using the top 500 genes with most biological overdispersion as estimated by the scran package.

**Code availability**. All R code is provided as Supplementary Software and is available from https://github.com/PMBio/scNMT-seq/

**Data availability**. Raw sequencing data together with processed files (RNA counts, CpG methylation reports, GpC accessibility reports) are available in the Gene Expression Omnibus under accession GSE109262. All other data are available from the authors upon reasonable request.

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

## Acknowledgements

We thank K. Tabbada and C. Murnane of the Babraham Next Generation Sequencing Facility for assistance with Illumina sequencing and R. Roberts of the Babraham Flow Cytometry Core Facility for assistance with FACS. W.R. was supported by the Biotechnology and Biological Sciences Research Council (BBSRC), the Wellcome Trust, the EU Blueprint and EpiGeneSys. G.K. was supported by the BBSRC and the Medical Research Council (MRC). O.S. is supported by the European Molecular Biology Laboratory (EMBL), the Wellcome Trust and the EU. C.-A.K. is supported in part by the EPSRC Centre for Doctoral Training in Data Science (grant EP/L016427/1) and the University of Edinburgh.

## Author contributions

S.J.C. conceived the method. S.J.C. and W.R. conceived the project. S.J.C., T.M.S., H.J.L. and C.A. performed experiments. R.A., S.J.C. and C.-A.K. performed statistical analysis. F.K. processed and managed sequencing data. S.J.C., R.A., J.C.M, O.S., W.R. interpreted results and drafted the manuscript. G.S., G.K., J.C.M, O.S., and W.R. supervised the project.

## Additional information

**Competing interests:** W.R. is a consultant and shareholder of Cambridge Epigenetix. The remaining authors declare no competing financial interests.

