## [Peer Review File · Nature Communications]

REVIEWERS' COMMENTS:

Reviewer #1 (Remarks to the Author):

The authors clarified several concerns of mine and the other reviewer in the revised manuscript. Both this reviewer and the other reviewer had expected a greater correlation between enhancer accessibility and gene expression. However, the authors have now provided additional analysis of publicly available bulk cell data on this relationship and found that there is no apparent correlation between enhancer accessibility and target gene expression even in bulk cells. If this is true, it would change the thoughts of field. Thus the authors should include this result in the final manuscript.

Reviewer #2 (Remarks to the Author):

In the revised manuscript and response to reviewer's comments, the authors perform a number of systematic assessments of correlations and variation in their data. These additional analyses add to the manuscript and directly address the majority of concerns regarding correlations between the three layers of omics data. While the accessibility / transcription correlations are not there, the authors have performed every reasonable test and comparison to other bulk data to suggest that their observations are valid. This is also backed up by their observed correlations between the other two layer comparisons that are concordant with expectations. The authors also perform a number of analyses to break down the variability of their data which suggests that technical variables are not likely responsible. Overall, I believe the revised manuscript is suitable for publication in its revised form.

Response to reviewers comments

Reviewer #1 (Remarks to the Author):

The authors clarified several concerns of mine and the other reviewer in the revised manuscript. Both this reviewer and the other reviewer had expected a greater correlation between enhancer accessibility and gene expression. However, the authors have now provided additional analysis of publicly available bulk cell data on this relationship and found that there is no apparent correlation between enhancer accessibility and target gene expression even in bulk cells. If this is true, it would change the thoughts of field. Thus the authors should include this result in the final manuscript.

This result is included in the main part of the manuscript in Figure 2 and we have updated the text to highlight the lack of correlation in active enhancers.

Reviewer #2 (Remarks to the Author):

In the revised manuscript and response to reviewer's comments, the authors perform a number of systematic assessments of correlations and variation in their data. These additional analyses add to the manuscript and directly address the majority of concerns regarding correlations between the three layers of omics data. While the accessibility / transcription correlations are not there, the authors have performed every reasonable test and comparison to other bulk data to suggest that their observations are valid. This is also backed up by their observed correlations between the other two layer comparisons that are concordant with expectations. The authors also perform a number of analyses to break down the variability of their data which suggests that technical variables are not likely responsible. Overall, I believe the revised manuscript is suitable for publication in its revised form.